# Anatomical and Simulation Studies Based on Three-Dimensional-Computed Tomography Image Reconstruction of Femoral Offset

**DOI:** 10.3390/diagnostics13081434

**Published:** 2023-04-16

**Authors:** Tomohiro Shimizu, Daisuke Takahashi, Hotaka Ishizu, Shunichi Yokota, Yoshihiro Hasebe, Keita Uetsuki, Norimasa Iwasaki

**Affiliations:** 1Department of Orthopaedic Surgery, Faculty of Medicine and Graduate School of Medicine, Hokkaido University, Sapporo 060-8638, Japan; 2R&D Center, Teijin Nakashima Medical Co., Ltd., Okayama 701-1221, Japan

**Keywords:** total hip arthroplasty, femoral offset, three-dimensional, Japanese population

## Abstract

Although the hip joint morphology varies by race, few studies have investigated the associations between two-dimensional (2D) and three-dimensional (3D) morphologies. This study aimed to use computed tomography simulation data and radiographic (2D) data to clarify the 3D length of offset, 3D changes in the hip center of rotation, and femoral offset as well as investigate the anatomical parameters associated with the 3D length and changes. Sixty-six Japanese patients with a normal femoral head shape on the contralateral side were selected. In addition to radiographic femoral, acetabular, and global offsets, 3D femoral and cup offsets were investigated using commercial software. Our findings revealed that the mean 3D femoral and cup offsets were 40.0 mm and 45.5 mm, respectively; both were distributed around the mean values. The difference between the 3D femoral and cup offsets (i.e., 5 mm) was associated with the 2D acetabular offset. The 3D femoral offset was associated with the body length. In conclusion, these findings can be applied to the design of better ethnic-specific stem designs and can help physicians achieve more accurate preoperative diagnoses.

## 1. Introduction

Total hip arthroplasty (THA) is one of the most performed and successful surgical treatments for patients with hip osteoarthritis [1]. Implant installation location and angle, offset, and limb length discrepancy are important perioperative factors [2,3,4]. While the increase in offset reduces the resultant force on the hip joint and increases the dislocation resistance, excessive offset extension causes greater trochanteric pain and a limited range of motion [5,6]. Additionally, several studies have revealed that offset could affect soft-tissue tension and the abductor muscle [2,7,8,9]. Therefore, a femoral implant for THA must effectively restore the biomechanics of the proximal femur and hip joint.

Although non-cemented THA has predominated worldwide, the durability and usefulness of cement stems have been reassessed recently [10,11]. One of the advantages of a cemented stem is its flexibility in terms of implant positioning, such as the depth of insertion and anteversion, to maximize the restoration of leg length or balancing of the soft tissue [12]. In particular, collarless polished stem designs can be inserted with flexibility. To achieve a reconstruction that is close to the normal hip joint, it is important to understand the anatomy and select the appropriate implant design. However, although the hip joint morphology varies by race [13], only a few studies have compared it between Asian and Western populations [14,15,16].

Considering that preoperative templating of femoral components on plain radiographs (two-dimensional [2D]) does not have a very high match rate with the stem size actually used [17], three-dimensional (3D) morphological characteristic information is thought to be more useful for achieving successful THA. Additionally, 3D changes in the hip rotation center (HRC) and femoral offset (FO) in THA [18] are also important and not fully understood when planning THA, including implant selection. Therefore, this study’s aims were as follows: (1) to clarify the 3D length of offset, 3D changes in the HRC, and the FO and (2) to investigate the anatomical parameters associated with the 3D length and changes using computed tomography (CT) simulation data and radiographic (2D) data.

## 2. Materials and Methods

This study has been approved by the Institutional Review Board of the authors. All patients gave informed consent before enrolment. Among the patients who underwent hip surgery at our hospital between January 2013 and December 2016, 66 patients whose femoral heads were kept spherical on the contralateral side were selected (28 men and 38 women). The exclusion criteria were as follows: (1) age < 19 years, (2) deformation of the femoral head, and (3) dysplasia (central edge [CE] angle [19] < 20 degrees). The mean age of the patients at observation was 55.4 years (range, 20–83 years; Table 1). The mean body length and body mass index (BMI) was 161.0 cm and 23.1 kg/m^2^, respectively.

Radiographs were obtained using Siebenrock’s standardized technique [20]. We evaluated the CE angle, FO (length of the perpendicular line between the HRC and femoral axis), acetabular offset (AO; length of the perpendicular line between the HRC and the vertical line across the inferior edge of the teardrop center), and global offset (GO; sum of the AO and FO; Figure 1) [21]. All digital measurements and calculations were performed using the Centricity™ Web-J 3.0 HD software (GE Healthcare Japan, Tokyo, Japan). Measurements were performed twice, with a 3-month interval, by the two first authors (T.S. and H.I.). The intra- and inter-class correlation coefficients were excellent (0.962 and 0.894, respectively; *p* < 0.001 for both).

We used a high-resolution (pixel matrix, 512 × 512) helical CT scanner (CT High Speed Advantage; GE Medical Systems, Milwaukee, WI, USA) to obtain axial images of the bilateral hips. The slice thickness and interval were set to 1 mm each, and the table speed was set to 1 mm/s. All CT slices were saved in the Digital Imaging and Communications in Medicine format.

To investigate the FO, the femoral local coordinate system was defined using bony landmarks on the 3D surface models following the International Society of Biomechanics recommendation [22]. Using commercial software (Materialise 3-matic, Materialise, Leuven, Belgium), the center of the femoral head was defined. The line connecting the midpoints of the femoral shaft was defined as the proximal femoral bone axis (Figure 2A).

For simulated implantation of the acetabular implant, the pelvic local coordinate system was defined using bony landmarks on the 3D surface models of the anterior pelvic plane [23]. Using commercial software (Mimics, Materialise; Leuven, Belgium), the maximum distance of the outer edge of the acetabulum was measured in the axial plane, and the acetabular cup size was determined (Figure 2B). A circle simulating a cup was positioned flush to the true floor of the acetabulum, parallel to the transverse acetabular ligament, as a landmark for acetabular anteversion (Figure 2C) [24].

Next, 3D matching of the pelvis and femur with the center of the acetabular and femoral head was performed using commercial software (Bone Simulator, ORTHREE, Osaka, Japan; Figure 3A). After 3D matching, the 3D center of the proximal femoral shaft (reference point), the center of the sphere approximated to the femoral head (FHC), and the simulated ball of cup (CC) were projected vertically (Figure 3B). Using another commercial software (Materialise Matic; Materialise, Leuven, Belgium), the distance between the reference point and FHC was defined as the 3D FO and the distance between the reference point and CC was defined as the 3D cup offset. The distance and angle of the cup-femoral center were also investigated.

The characteristics of the cases were compared between men and women and between the 3D FO and 3D cup offset using an independent Student’s *t*-test. Correlations regarding radiological parameters were analyzed using Pearson’s product–moment correlation coefficients. All statistical analyses were performed using IBM SPSS version software (SPSS Inc., Chicago, IL, USA), and the statistical significance was set at *p* < 0.05.

## 3. Results

Table 1 summarizes the biographic and radiographic parameters of the included patients. The mean body length of men was significantly higher than that of women (*p* < 0.001). There were no significant differences in the mean age and body BMI between men and women. The mean CE angle of women was significantly lower than that of men (*p* = 0.015). The mean GO, AO, and FO of men were significantly higher than those of women (*p* < 0.001, *p* < 0.001, and *p* = 0.005, respectively). The GO, AO, and FO were significantly associated with the body length (*p* < 0.001, *p* < 0.001, and *p* = 0.021, respectively; Figure 4).

Figure 5A shows the distribution of the 3D FO and the cup offset. The mean 3D FO and 3D cup offsets were 40.0 mm (range, 30.9–52.6 mm) and 45.5 mm (range, 36.8–60.3 mm), respectively. Both 3D offsets were distributed around the mean values. The mean 3D cup offset was significantly larger than the mean 3D FO (*p* < 0.001). There was a significant association between the 3D FO and 3D cup offsets (R = 0.915, *p* < 0.001; Figure 5B). The mean 3D FO, cup offset, difference between 3D FO and cup offset, and femoral anteversion were significantly larger in men than in women (Table 2). No significant differences were observed between the distance and angle of the cup-femoral center.

As seen in Figure 6, the difference between the 3D FO and cup offset was associated with the 2D GO (R = 0.310, *p* = 0.011), AO (R = 0.640, *p* < 0.001), and CE angle (R = 0.259, *p* = 0.036). The 3D FO was associated with the body length (R = 0.397, *p* = 0.001; Figure 7A). As expected, the 2D FO was associated with the 3D FO (R = 0.728, *p* < 0.001; Figure 7B). Additionally, the difference between the 3D and 2D FO was associated with femoral anteversion (R = 0.343, *p* = 0.005; Figure 7C).

## 4. Discussion

A recent registry study revealed that implant choice is a factor associated with implant survival following THA [25]. Therefore, the selection of an implant that matches the anatomical features seems to be key to the safe placement and durability of the prosthesis. Additionally, to optimize function, hip mechanics should be restored to as near normal as possible. Determining the appropriate offset in THA is important from the perspective of the effect on the joint forces and range of motion applied to the hip by the acquisition of dislocation resistance and definition of the lever arm of the abductor muscle. In addition to FO and AO, the GO (which is the distance from the pubic symphysis to the femoral axis) should be evaluated for the hip offset. In the past, preoperative and postoperative evaluations were often performed using 2D frontal radiographic images of the hip joint. However, in recent years, 3D evaluation using CT has improved our understanding of the anatomy and enabled detailed preoperative and postoperative planning and evaluation. Factors that determine the offset include the thickness of the acetabular base, the presence or absence of a double floor, joint congruity, deformation of the femoral head, and anteversion and cervical angles of the femur. The position of the hip center on the affected and healthy sides should be accurately evaluated in two and three dimensions using simple radiography and CT, and the offset should be determined from the viewpoint of biomechanics (kinematics and kinetics) during movement.

This study primarily aimed to investigate the 3D FO length in the Japanese population, in order to determine its potential for use in the selection or manufacture of appropriate implants that match the anatomical features. The mean 3D FO length in this study was 40.0 mm (men, 42.1 mm; women, 38.4 mm), which was smaller than that reported previously in the Western population [26,27]. As expected, the findings suggest that the FO may tend to be smaller in the Asian population than in the Western population. Although Takamatsu et al. reported that there were no correlations between the FO length and the anatomical morphologies of the pelvis [15], this study found that there were significant associations between the body length and the 2D and 3D FO. Considering that the difference between the 3D-2D FO was associated with femoral anteversion, 3D evaluation is important and necessary for the reconstruction of accurate FO.

This simulation study found that the 3D cup offset was larger than the 3D FO, and the mean difference was 5.5 mm. This result was consistent with that of a previous CT study, which showed that medialization of the HCR was >5 mm in almost half of the cases when the conventional cup insert technique was used [28]. Clinically, this discrepancy may be important for patients who undergo THA following hemiarthroplasty for displaced femoral neck fractures [29]. To decrease the GO due to medialization of the hip center, complementing with the FO may be effective. In addition, the height of the inner head or offset liner could adjust the GO; however, there could be issues in considering the leg length discrepancy or linear wear [30]. Therefore, implants with appropriate offset length variations are useful for the reconstruction of GO.

This study also found that the difference in the 3D cup-femoral offset was strongly associated with the 2D AO. These findings suggest that a large AO could lead to medialization of the hip center during THA and should be considered when selecting a high offset size. Clinically, AO varies widely among individuals, which should not be ignored [28]. Because the difference between a standard and a high offset stem was limited, Meermans et al. reported that the loss in offset on the acetabular side cannot always be compensated on the femoral side and recommended peripheral reaming rather than standard reaming [31]. However, considering that the maximum difference in the 3D cup-femoral offset in the current study was approximately 10 mm and smaller than that in the Western study [31], stem offset selection could compensate for the medialization of the hip center in this population.

This current simulation study had some limitations. First, the cup was positioned flush with the true floor of the acetabulum. Because the AO varies widely among individuals, simulation performed using the anatomical cup implantation [28] or peripheral reaming technique [31] makes it difficult to unify the study conditions. Second, this study included only patients with unilateral hip disease. Although the study of a healthy control population is ideal, it is difficult due to radiation exposure. Third, it had a relatively small sample size, considering age and sex variations. Therefore, a further study on a larger healthy population may be necessary. However, a clear trend in the distribution of the offsets in the Japanese population was obtained in this study.

## 5. Conclusions

This study showed that the mean FO was 40 mm; the difference between the cup-femoral offset was 5 mm and was associated with the AO in the Japanese population. These findings can be applied to design better ethnic-specific stem designs and preoperative diagnosis.

## Figures and Tables

**Figure 1 diagnostics-13-01434-f001:**
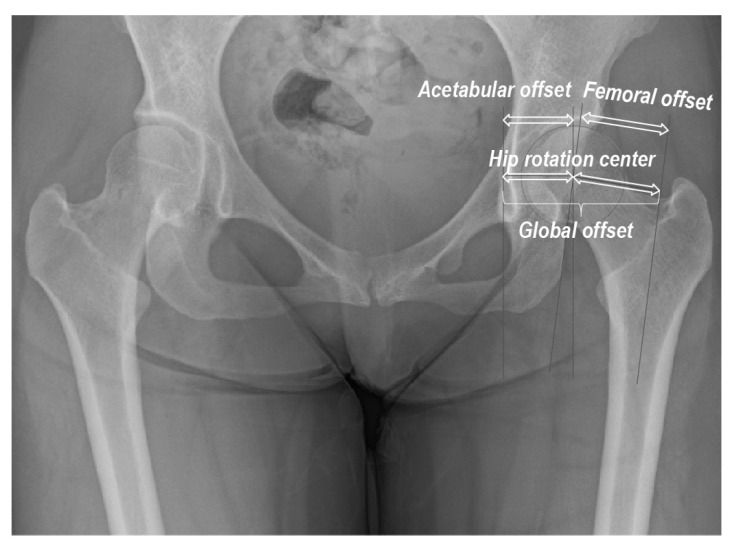
Radiographic hip morphometric parameters analyzed in this study.

**Figure 2 diagnostics-13-01434-f002:**
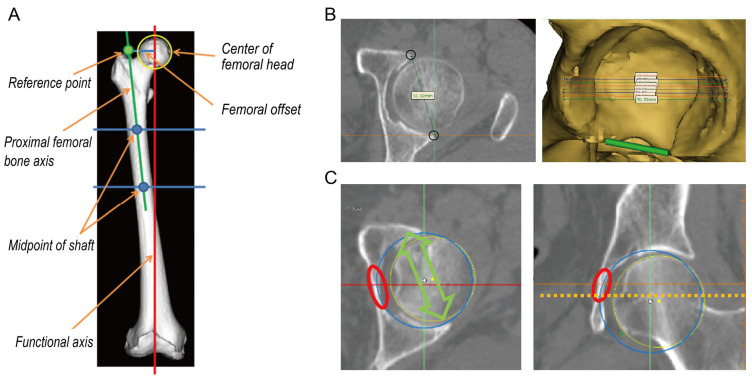
(**A**) Three-dimensional femoral bone model; (**B**) Left: axial slice of the maximum distance of the outer edge of the acetabulum; Right: three-dimensional acetabular bone model; (**C**) a blue circle simulating a cup is positioned flush to the true floor of the acetabulum Right: axial view; Left: coronal view.

**Figure 3 diagnostics-13-01434-f003:**
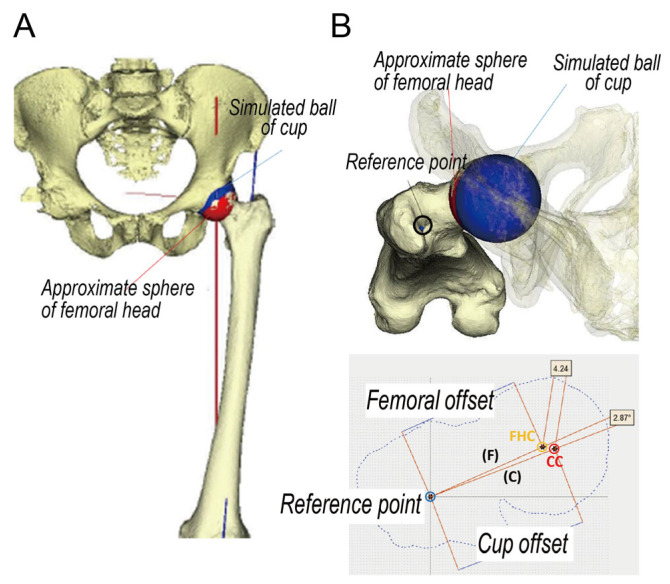
(**A**) Three-dimensional matching bone model. (**B**) Projection image of a three-dimensional matching bone model. F: femoral offset, C: cup offset.

**Figure 4 diagnostics-13-01434-f004:**
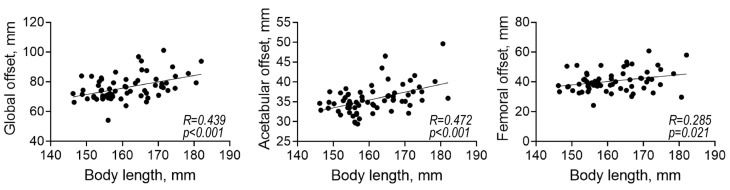
Correlations among the radiographic parameters and body length.

**Figure 5 diagnostics-13-01434-f005:**
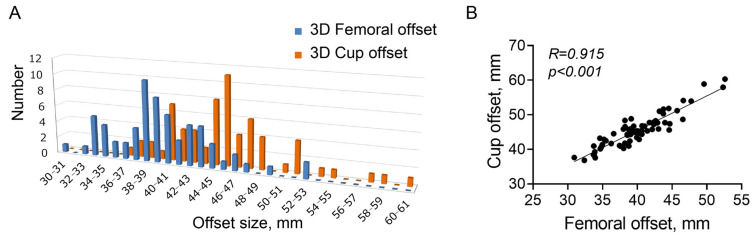
(**A**) Distribution of the three-dimensional femoral and cup offsets. (**B**) Correlation between the three-dimensional cup and femoral offsets.

**Figure 6 diagnostics-13-01434-f006:**
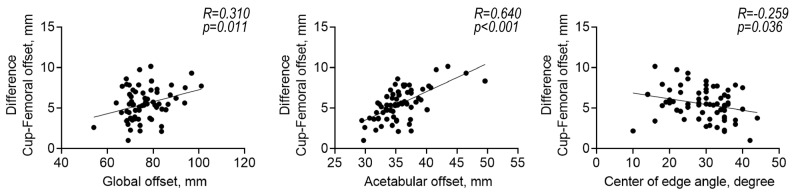
Correlations among the radiographic parameters and the difference between the cup and femoral offsets.

**Figure 7 diagnostics-13-01434-f007:**
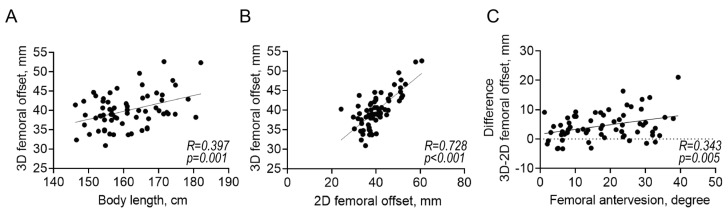
Correlations between (**A**) the body length and three-dimensional femoral offset, (**B**) two-dimensional and three-dimensional offset, and (**C**) femoral anteversion and difference three-two-dimensional femoral offset.

**Table 1 diagnostics-13-01434-t001:** Patient characteristics and radiographic parameters.

	Men	Women	*p*-Value
Number	28	38	
Age (year)	52.5 (15.5)	58.0 (15.7)	0.159
Body length (cm)	169.4 (5.6)	154.9 (4.0)	<0.001
Body mass index (kg/m^2^)	23.5 (3.2)	23.6 (4.5)	0.902
Center of edge angle, (°)	31.9 (5.3)	28.1 (6.8)	0.015
Global offset (mm)	80.9 (8.8)	72.6 (5.9)	<0.001
Acetabular offset (mm)	37.7 (4.0)	34.2 (2.4)	<0.001
Femoral offset (mm)	43.2 (8.3)	38.4 (5.0)	0.005

**Table 2 diagnostics-13-01434-t002:** Three dimensional parameters.

	Men	Women	*p*-Value
Number	28	38	
Femoral offset, mm	42.1 (4.8)	38.4 (3.6)	<0.001
Cup offset, mm	48.4 (5.1)	43.3 (3.4)	<0.001
Cup-femoral offset, mm	6.3 (1.9)	4.9 (1.8)	0.002
Distance cup-head center, mm	8.1 (3.0)	7.2 (2.3)	0.146
Angle cup-head center, °	6.2 (4.3)	7.1 (3.4)	0.335
Femoral anteversion, °	13.2 (9.6)	21.2 (9.9)	0.002

## Data Availability

The data presented in this study are available on request from the corresponding author.

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
