# Peer review of "Anatomical and Simulation Studies Based on Three-Dimensional-Computed Tomography Image Reconstruction of Femoral Offset"

_diagnostics, 2023, doi:10.3390/diagnostics13081434_

Round 1
Reviewer 1 Report
General comments
I understood the authors found the standard values such as 3D FO and 3D cup offset in Japanese population. The authors also referred the clinical implications of them and those results were right, however, I did not understand the meaning of the values. I think the authors describe only the standard values in Japanese population. They should describe the clinical implication of each finding.
Specific comments
- In methos section, the authors used the parameters for length, such as AO, FO, GO. How about adjustment by body size ? The difference of body height between men and women was 15cm, but those of these parameters were just 3-4mm.
- Figure 1 and 2 , the measurement method of FO was different between 2D and 3D. It should be unified.
- In line 106, what was their “reference point” ?
- In line 177-179, how do the surgeons use their results clinically ? I did not understand this sentence.
- In line 201, they argued small sample, but clear trend. Is it valid ? Power value was 0.5 in t-test and 0.7 in correlation, I speculated.
Reviewer 2 Report
The study is well-designed and offers important information regarding the total hip implant design in Japanese patients.
Author Response
Thank you for your favorable comments.
Reviewer 3 Report
The proposed study is certainly very interesting, and I congratulate with the Authors for the clear exposition. As you have correctly expounded, femoral THA implantation must effectively restore the biomechanics of the proximal femur and hip joint. Therefore, the operative strategy must be planned in the direction of restoring the anatomical conditions as close to the physiological condition as possible, so as to have a balance between joint work and muscle work. It is evident how a greater off-set results in a greater lever arm of the hip abductor muscles and an advantage in both mechanical and muscle strength, leading to an increase in joint range especially of hip abduction, a decrease in the incidence of impingement of the femur with the pelvis, and a greater stability of the prosthetic implant.
I would ask you to specify what, in your experience, are the key factors for safe placement and durability of the prosthesis and whether you feel that a biomechanical compromise should be made in relation to joint stability and motion that can ensure sufficient off-set and abductor lever arm values.
I am sure that by making the minor additions required the impact of this work could be interesting.
In conclusion, the study certainly has interesting points. The purpose is clear and respected. The main question addressed by the research is clear and entirely agreeable. I believe that the information provided is sufficient and represents useful elements to encourage the development of new scientific work.
Author Response
Thank you for your favorable comments. We believe that the key factor for the safe placement and durability of the prosthesis is the selection an implant that matches the anatomical features. Because we believe that hip mechanics should be restored to an as near normal state as possible, we do not agree with a biomechanical compromise. We revised the Discussion to reflect this
Page 6, Lines 161–164:
“A recent registry study revealed the implant choice as a factor associated with implant survival following THA [25]. Therefore, selection of an implant that matches the anatomical features seems to be key to the safe placement and durability of the prothesis. Additionally, to optimize function, hip mechanics should be restored to as near normal as possible.”
Round 2
Reviewer 1 Report
I think the authors did not appropriately reply to the reviewer's question.